# Competitive Advantage in the World of Wine—An Analysis of Differentiation Strategies Developed by Sectoral Brands in the Global Market

**DOI:** 10.3390/foods14111858

**Published:** 2025-05-23

**Authors:** Daniel Marian Micu, Georgiana Armenița Arghiroiu, Ștefan Micu, Silviu Beciu

**Affiliations:** 1Department of Management and Marketing, University of Agricultural Sciences and Veterinary Medicine of Bucharest, 59 Marasti Blvd, District 1, 11464 Bucharest, Romania; daniel-marian.micu@doctorat.usamv.ro; 2Department of Finance, London School of Economics and Political Sciences, Houghton Street, London WC2A 2AE, UK; s.micu@lse.ac.uk

**Keywords:** global wine market, global wine industry, competitive advantage, differentiation, cost leadership, sectoral brand, differentiation strategies, differentiation attributes, cluster analysis

## Abstract

This study aims to analyze the differentiation strategies developed by sectoral brands in the global wine industry and how these strategies interrelate. A sectoral brand is defined as having a distinctive brand name accompanied by a visual identity, with or without a slogan. Through an analysis of thirty-three sectoral brands developed by wine-producing countries, seven clusters of differentiation strategies and three clusters of differentiation attributes were identified, using quantitative and qualitative methods. The findings highlight the alignment between the differentiation strategies employed by sectoral brands and the underlying theoretical concepts, as well as overlaps between differentiation strategies and specific attributes. The results identify unrevealed opportunities for wine-producing countries that have not yet developed sectoral brands. This study’s main contribution consists of the application of a cluster analysis approach, which enabled the identification and interpretation of relationships among sectoral wine brands based on their differentiation strategies. Accordingly, the research addresses a notable gap in the existing literature by providing an integrative perspective on how sectoral brands differentiate within the world wine market. The practical implications of this study include offering valuable guidance to countries currently lacking sectoral wine brands and presenting a structured framework to effectively leverage unique national attributes.

## 1. Introduction

The wine market and industry in the European Union (EU) and at a global level are at a crossroads [1,2]. A summary of the statistical data reveals the following situation: the global vineyard area experienced a slight decline in 2023 compared with 2022 (−0.5%); global wine production, according to preliminary data for 2024, reached 227 million hectoliters, marking a 3.4% decrease from the 2023 volume (which was already below the levels of the past 25 years); and global wine consumption dropped from 234 million hectoliters in 2021 to 221 million hectoliters in 2023, representing a −5.6% reduction (with 2023 global consumption being −7.92% lower than the average of the entire previous decade) [3]. The long-term downward trend in wine consumption can be explained by economic crises and shifts in consumer preferences, as traditional markets (Europe, China) continue to consume less wine while emerging markets fail to compensate for this decline. Globally, the main reasons behind the decrease in wine consumption are: (i) declining incomes, (ii) the adoption of a healthy lifestyle and increased awareness of the negative effects of alcohol on health, and (iii) changing consumer preferences toward craft beer, cocktails, and spirits [4]. 

In 2023, with a +1.97% increase compared with 2022, the growth rate of the average price per liter of exported wine appears to be slowing down, indicating a possible short-term stabilization. Meanwhile, after peaking in 2017 (110 mhl.), the global volume of wine exports has fluctuated and started to decline more sharply after 2021, reaching 99.3 mhl. in 2023 due to the global economic crisis, the impact of the pandemic, and changes in consumption habits [3]. On the other hand, the overall trend in export value has been one of steady growth. In 2022, wine export value reached EUR 37.7 billion, but this peak was followed by a slight decline in 2023 to EUR 36 billion, possibly due to post-pandemic consumption normalization and decreasing demand in certain markets [3].

In response to the challenges affecting the wine market, wine producers must adapt their business strategies, including production, commercial, and marketing strategies [5]. Some of these producers have chosen to focus on agro-ecological and sustainability challenges and the need to promote new grape varieties that are more resilient, require fewer treatments, and thus reduce operating costs to shift consumer perception [6,7]. On the other hand, many wine-producing countries have chosen to intensify their marketing programs as part of their efforts to counteract these negative developments [4]. In this context, and given the importance of the country of origin in wine purchasing decisions, numerous wine-producing countries have opted to create and develop a sectoral brand for their national wine industry as part of their efforts to promote exports in an extremely competitive global market [8,9,10,11,12].

The importance of the country of origin in wine trade is a subject that has been extensively analyzed both in academic literature and in current commercial practice. However, to date, there is a lack of research examining how sectoral brands within the global wine industry formulate their differentiation strategies, taking into account both their own resources and the strategic decisions of their competitors. This research addresses an existing gap by applying cluster analysis to uncover differentiation strategy patterns among sectoral brands, an approach seldom employed in preceding studies within the wine sector [13]. The wine industry provides an ideal context for studying differentiation strategies due to its strong dependence on intangible elements such as regional identity, terroir, cultural heritage, and consumer perceptions of quality [14]. Given the declining global wine consumption, increasingly intense competition, and fear for highly international trade protectionism, understanding sectoral branding strategies is essential for achieving sustainable competitive advantages. As a result, this research aims to address the subsequent questions: (Q1) What are the key differentiation strategies of sectoral brands in the global wine industry? (Q2) How does each of these sectoral brands relate to the others based on the chosen differentiation strategies?

This study investigates how differentiation strategies contribute to building competitive advantage for sectoral brands within the global wine industry. Drawing on existing theoretical frameworks, the research develops a methodology to analyze brand differentiation strategies. The findings offer practical implications for wine-producing countries, marketing and brand managers, and policymakers, highlighting effective approaches to strengthen market presence through targeted differentiation efforts.

## 2. Literature Review

### 2.1. Competitive Advantage

Competitive advantage is a central factor in organizational performance, especially in markets characterized by high competition [15]. After a period of rapid growth and prosperity, many organizations tend to underestimate the role of competitive advantage, focusing primarily on expansion policies, but the relevance of competitive advantage remains just as significant as it was in the early stages of organizational development [15]. The intensification of competition, on both local and global scales, has led to a decrease in the predictability of profitability in most industries. In this context, the central objective of an effective strategy must be the development and maintenance of a sustainable competitive advantage for the organization [16].

Competitive advantage is achieved when an organization creates or obtains a specific set of attributes or implements strategic actions that provide it with a superior advantage over competitors. There are two dominant theories in the early stages of competitive advantage research: the market-based view (MBV) and the resource-based view (RBV). Later, conceptual developments such as the knowledge-based view and the capability-based view emerged from the theoretical framework of RBV [17]. The concept of competitive advantage can be defined as an organization’s superiority in generating value for stakeholders or achieving higher profitability [16]. Competitive advantage is not a static or stable phenomenon but a dynamic one characterized by imbalance. It is driven by change and, once achieved, triggers a competitive process that, over time, leads to its erosion. Therefore, it is essential for organizations to continuously review and rebuild their competitive advantage to maintain their market position [16].

There are two fundamental ways in which an organization can achieve competitive advantage: cost leadership or differentiation [15]. These two fundamental types of competitive advantage, along with the range of activities an organization aims to apply them to, result in three generic strategies: cost leadership, differentiation, and focus. The focus strategy itself has two variations: cost focus and differentiation focus [15]. While the cost leadership strategy involves an organization striving to achieve the lowest cost structure in the entire industry in which it operates, thus securing a competitive advantage, the differentiation strategy aims to create a unique market position by offering distinct features valued by consumers. In this context, the organization identifies and capitalizes on one or more attributes perceived as essential by customers, thereby differentiating itself from competitors. This uniqueness allows the organization to charge a premium price, reflecting the perceived value by buyers [15]. In addition, a recent study reveals that the unique capabilities of an export-oriented organization, particularly informational, relational, and marketing capabilities, as well as differentiation and cost leadership strategies, provide a competitive advantage and enhance its performance in foreign markets [18]. 

Even though some researchers argue that low-cost and differentiation strategies can be combined and implemented by an organization simultaneously, and that this strategic combination has a significant impact on organizational performance compared with a single strategy [19], the differentiation strategy is the one that emerges as particularly relevant. It enables organizations to create unique market positions that enhance their competitiveness beyond cost advantages. Furthermore, recent research indicates that in the wine industry, it is advisable to choose one of the two strategies in order to avoid being stuck in the middle and diminishing brand equity [20].

### 2.2. Differentiation

Every good and service is unique in some way, and there is no truly generic commodity [21]. Differentiation is not limited to offering customers only what they expect but also includes products and services with features they have never even considered [21]. Differentiation is understood as the process of positioning products or services so that they are perceived as distinct from those of competitors, with branding representing the ultimate expression of this endeavor [22].

The process of differentiation relies on using non-price-related factors to reduce competitive intensity [23]; Kotler and Keller argue that an organization’s strategy should focus on creating a high degree of product differentiation compared to competitors so that consumers perceive them as superior, and a high level of differentiation leads to a decrease in direct competition [24].

Competitive differentiation has become critically important in an era of hyper-competitiveness [25], whereas a study on firms in the U.K. established that only the differentiation strategy has significant links to financial performance [26]. The differentiation strategy was identified and conceptualized as a key approach for SMEs aiming to successfully export to international markets [27]. Furthermore, a conclusion of their study states that differentiation strategies are the most suitable for international new ventures to achieve international performance [28].

In a consumer-centered approach, differentiation is defined as the process of designing and communicating distinct elements that set an organization’s offering apart from those of competitors [29]. Differentiation occurs at the perceptual level, in the consumer’s mind [29,30]. This entails that marketing strategies should focus on promoting a single product, highlighting a key benefit through a clear message [29].

Differentiation is a fundamental concept with historical roots, essential for understanding markets and modern marketing strategies [31]. Sharp and Dawes emphasize that differentiation makes a product desirable, determining its uniqueness and value, which in turn strengthens brand loyalty. This process has direct implications for profitability, consumer demand, the ability to charge premium prices, and reducing operational and marketing costs [31]. The authors conclude that the academic literature highlights two major effects of differentiation: (a) reducing the intensity of direct competition and (b) decreasing consumers’ price sensitivity. They also observe that the term differentiation is used in various ways, including meanings such as perceived superiority, a component of marketing communication, or an organization’s ability to offer something distinct from the competition [31].

Product or service differentiation is a distinctive characteristic of imperfect markets, where non-price strategies play a crucial role [32]. This process can involve tangible differences such as quality, reliability, performance, or design, but it can equally be based on intangible elements such as reputation and branding. This perspective aligns with the prevailing view in the literature, which suggests that developing and implementing an effective differentiation strategy requires the creation and strengthening of a strong brand [32].

A detailed and applied research approach to the strategic process of differentiation proposes a series of essential steps for its success [33]: (a) the differentiation strategy must be coherent from both a market (consumer) and industry (competition) perspective; (b) identifying a distinctive idea of differentiation that is authentic and originates within the organization; (c) basing the strategy on relevant credentials; (d) effectively communicating the strategy; and (e) ensuring its uniqueness. Additionally, the authors identify and analyze several possible differentiation strategies: (1) through a distinctive product (or brand) attribute that must be unique and relevant; (2) by emphasizing the brand’s market leadership position if such a position is held; (3) through the characteristics of the product or service production process; (4) by leveraging the brand’s heritage, which has contributed to its evolution; (5) through tradition, both at the brand level and in terms of country of origin; (6) by being the first brand launched on the market; (7) by positioning as the latest or most recent brand introduced; (8) by specializing in a particular field or market segment; and (9) through the preference expressed by a specific group of consumers toward the brand [33].

A fundamental connection exists between the differentiation strategy and the overall business strategy. Differentiation should not be seen as an end in itself but as a process focused on deeply understanding customers and better satisfying their needs. In this context, achieving a competitive advantage through differentiation becomes a central element of business strategy, contributing to strengthening the organization’s market position [16,34].

The importance of branding and differentiation in the wine industry is highlighted by Resnik and Lorenzon et al. [35,36]. Modern wine marketing demands a fundamental reassessment of business strategies to effectively expand market share and bolster global reputation [37]. A wine brand is more easily recognized by consumers due to its consistency in quality and taste. Building a strong brand in the wine sector can be achieved by associating with: (a) a specific place, such as a country, region, or terroir; (b) a distinct grape variety; and (c) a lifestyle image, which contributes to strengthening the brand’s identity and appeal [35]. Even at a regional level, the success of wine-producing companies is more closely associated with a differentiation strategy rather than with cost leadership [38].

Branding’s significance in the wine industry is also highlighted by Harvey et al. [39], who state that global wine markets are dynamic, fluctuating, and highly competitive. This is partly because wine differs significantly from other agricultural products. Consumers seek information about where, when, and how the wine was produced, and these elements are major criteria in their purchasing decisions [39]. The authors also emphasize that wine is distinguished by its identity (a blend of brand, heritage, and terroir), which grants specific wines and wine regions a competitive advantage. In their article [40], which focuses on wine brand positioning configuration among winery brands in Germany, Dressler and Paunovic start from the distinction between the concepts of brand identity and brand image to create world–price clusters and provide insights into communication and pricing opportunities for these concepts.

Within the broader framework of differentiation strategies, sectoral branding has become an increasingly effective instrument for emphasizing distinctive regional and national attributes, especially in industries such as wine, where product origin and cultural identity significantly shape consumer preferences and perceptions.

### 2.3. Sectoral Brands

Based on the definition provided by the American Market Association [41], which states that a brand is any distinctive feature like a name, term, design, or symbol that identifies goods or services, Stanton et al. [42] expand on this and define a sectoral brand as “a group of products from a specific sector of a country, aimed at identifying and differentiating them from products in the same category from other countries”. A sectoral brand functions by leveraging both competitive and comparative advantages and is closely tied to a geographical concentration, typically formed around a specific product or industrial sector [43]. Sectoral branding involves the collaboration of entrepreneurs within the same industry, either independently or with the backing of the country brand. Each sectoral brand can represent a single sector, with its own distinct identity [44]. The importance of country of origin and, consequently, sectoral branding in the wine industry is also reflected in the number of scientific studies conducted over the past three decades in wine marketing research [45].

Sectoral wine brands are particularly relevant because they promote the entire industry, both domestically and, more importantly, internationally. These brands can be considered intangible resources that must be developed internally by the respective sector. Therefore, those managing these brands must analyze the industry from within and identify elements that are valuable, rare, and costly to imitate, leveraging them to achieve a sustainable competitive advantage [46,47].

In the context of international trade, when competitors can offer a quasi-identical substitute product in terms of price and quality, consumers begin to look at other attributes that help them differentiate between brands from different countries [48,49]. As a result, the importance of intangible resources such as sectoral brands increases, and these brands become increasingly difficult for competitors to imitate, as their construction involves unique elements developed over time by an organization with its own culture [50]. Empirical evidence has shown that regional and sectoral wine brand equity significantly influences consumer perceived quality and preferences [51,52,53]. Even more, in the case of wine, data-driven research has demonstrated that country of origin influences to varying extents the different dimensions of brand equity, including brand loyalty, perceived quality, brand awareness, and brand associations [54].

However, to prevent imitation by competitors and protect the sectoral brand from its natural depreciation, an organization must continuously invest in performance, innovation, design, and style [55]. But wine producers should not rely solely on sectoral or regional brands. They must develop strong individual brands in order to meet the increasingly demanding expectations of consumers in an increasingly competitive global market. Producers who rely exclusively on sectoral or regional brand reputation will continue to target less discerning consumers, who depend on more diffused quality signals, within a less mature market context [56]. Although sectoral, regional, and appellation-based wine brands play a critical role, they alone are not sufficient to ensure successful market outcomes [57]. The success of sectoral (territorial) wine brands is largely driven by strategic brand management, effective coordination and co-branding among individual producers, the development of a compelling shared brand narrative, and authentic engagement with local traditions and community values to enhance brand equity and consumer loyalty [58].

The structure of the study reflects a coherent progression from conceptual grounding to practical application. The definition of sectoral brands establishes the foundation for the analysis, which is then operationalized through a mixed-method approach involving both quantitative and qualitative techniques. This methodological framework supports the identification of strategic and attribute-based clusters, revealing patterns of differentiation among the examined sectoral brands. The integration of cluster analysis not only facilitates the interpretation of complex relationships but also bridges the gap in the literature by offering a comprehensive perspective on differentiation in the global wine industry. Together, these sections contribute to a unified narrative that advances both academic understanding and practical strategy development.

## 3. Methodology

From a methodological perspective, the research was deliberately not restricted to the main wine-producing countries (in terms of production volume). In the initial stage of the research, a database was developed with sixty countries (South Africa, Argentina, Armenia, Australia, Austria, Belgium, Bosnia and Herzegovina, Brazil, Bulgaria, Canada, Czech Republic, Chile, China, Cyprus, South Korea, Costa Rica, Croatia, Denmark, Switzerland, Estonia, France, Georgia, Germany, Greece, India, Ireland, Israel, Italy, Japan, Kazakhstan, Latvia, Lebanon, Lichtenstein, Lithuania, Luxembourg, North Macedonia, Great Britain, Mexico, Moldova, Norway, New Zealand, the Netherlands, Paraguay, Peru, Poland, Portugal, Romania, Russia, Serbia, Singapore, Slovakia, Slovenia, Spain, the USA, Sweden, Turkey, Ukraine, Hungary, Uruguay, and Venezuela). These countries were included based on their Organisation Internationale de la Vigne et du Vin (OIV) wine production statistics and/or their participation with national pavilions at Pro Wein Düsseldorf 2024, the world’s most important wine trade fair.

In the second stage, the research investigated which of these countries had developed a sectoral brand for their national wine industry. The key criteria for analyzing and determining the presence of a sectoral brand in the global wine industry were: (a) the existence of a distinct brand name, different from the country’s name; and (b) the presence of a visual identity (logo), with or without an accompanying slogan. After applying these criteria, the initial database was reduced from sixty to thirty-three countries, namely: South Africa, Argentina, Armenia, Australia, Austria, Belgium, Brazil, Bulgaria, Canada, Chile, China, Cyprus, Croatia, Switzerland, Estonia, Georgia, Germany, Greece, Israel, Lebanon, Lichtenstein, Lithuania, North Macedonia, Great Britain, Moldova, New Zealand, Portugal, Slovenia, Spain, Turkey, Ukraine, Hungary, and Uruguay.

Countries excluded from this refined database included Italy, France, and the USA, which, due to their global prestige in the wine industry, do not develop sectoral brands but rather focus on regional wine brands (e.g., Tuscany, Umbria, Sicily, Bordeaux, Burgundy, California, Washington, Oregon, etc.). Other countries were excluded because they had not yet developed a sectoral brand (e.g., India, The Netherlands, Paraguay, Romania, etc.).

The final sample was analyzed both quantitatively and qualitatively. For the thirty-three countries, official sectoral brand websites or websites that promote the wine industry were considered (Appendix A). The purpose was to extract keywords, expressions, slogans, and other marketing and branding elements to identify differentiation strategies and the attributes used for differentiation.

The identification criteria for these differentiation strategies were based on the differentiation methods suggested by Gwin C.F. and Gwin C.R. [59] but primarily by Trout and Rivkin [33], which then became the analysis variables. These differentiation methods include: (1) brand attribute-based differentiation, where the attribute must be unique and relevant; (2) brand leadership-based differentiation, if such leadership exists; (3) product manufacturing (“how it is made”)-based differentiation; (4) heritage-based differentiation, where the brand’s historical background contributes to its evolution; (5) tradition-based differentiation, considered at both the brand and country level; (6) first brand on the market differentiation; (7) latest brand on the market differentiation; (8) specialist brand differentiation, where the brand is recognized for expertise in a specific domain/market segment; and (9) consumer preference-based differentiation, where a specific consumer segment prefers a particular brand [33].

The keywords, attributes, expressions, and slogans identified were distinctly grouped based on each differentiation method. In cases where a clear, unique assignment was not possible, the same word, attribute, expression, or slogan was classified under multiple differentiation methods. The results of this process are also presented in Appendix A.

No specific software was used in this stage; instead, the data extraction was performed manually. The identified words, attributes, expressions, and slogans are nominal data by nature. For statistical analysis and interpretation, these were coded using binary values (0 for absent, 1 for present). Similarly, after identification, the differentiation attributes were also coded.

In the next stage of the research, the objective was to understand: (1) how each sectoral brand in the global wine industry relates to others based on the differentiation strategies used; and (2) how sectoral brands that differentiate (also) through an attribute position themselves in relation to the others. After coding the nominal data by assigning binary values, a cluster analysis was applied, involving the use of classification algorithms that allow for grouping objects into homogeneous clusters. However, it is important to note that the use of different algorithms may lead to different classifications [60]. In this case, two types of classification algorithms were used: hierarchical clustering and k-means clustering, with the results visualized using dendrograms and graphs.

As an initial step, a hierarchical clustering algorithm with a single linkage was applied to generate a dendrogram. The formation of clusters was based on the calculation of the Hamming distance matrix between any two objects in the binary dataset. The Hamming distance between two equal-length strings is the number of positions at which their corresponding symbols differ. The clustering algorithm used this precomputed distance matrix and proceeded iteratively, merging the closest clusters at each step based on the single linkage principle. A single linkage defines the distance between two clusters as the minimum distance between any two objects, one from each cluster, using values from the initial Hamming distance matrix. The final dendrogram illustrates similarity (computed as 1—Hamming distance), visually representing the hierarchical structure of relationships between all analyzed variables. This visualization enables a clear understanding of how variables naturally group, with the dendrogram branches indicating when clusters merge and their height representing the degree of similarity between groupings. In the next step, the k-means algorithm was applied, using Hamming distance to measure similarity between elements. Initially, k cluster centers were randomly selected, and each data point was assigned to the cluster with the minimum Hamming distance. The cluster centers were then recalibrated by computing the majority value at each position of the assigned vectors. This process was repeated iteratively until the clusters stabilized. The optimal number of clusters was determined using the silhouette score method.

All statistical operations described above were performed using Python 3.8.10, utilizing the SciPy 1.7.3, Scikit-learn 1.2, Matplotlib 3.1.1 and Seaborn 0.13.2 libraries. Through multiple iterations, the results revealed seven clusters of sectoral brands in the global wine industry based on differentiation strategies and three clusters based on the specific attribute(s) utilized in these strategies. A flowchart outlining the steps of the research methodology is presented in the figure below (Figure 1).

## 4. Results and Discussion

An initial analysis of the collected data focuses on the number of differentiation strategies employed by sectoral brands in the global wine industry (Figure 2).

Seven of the analyzed sectoral brands (21.2%) have chosen a single differentiation strategy. These correspond to the wine industry in the following countries: Bulgaria, China, Estonia, Lichtenstein, Lithuania, Spain, and Uruguay. The most common approach is the use of two differentiation strategies, observed in sectoral brands from nine countries (27.3% of the total): Armenia, Australia, Austria, Belgium, Brazil, Chile, Cyprus, Great Britain, and Ukraine. These countries appear to adopt a more diversified approach, attempting to combine two elements specific to their respective wine industries. The use of three differentiation strategies is also relatively frequent, adopted by sectoral brands in eight countries (24.2% of the total): Canada, Croatia, Germany, Israel, North Macedonia, New Zealand, Slovenia, and Turkey. This situation can be viewed as a more complex approach, potentially reflecting either a simultaneous presence in multiple competitive markets or a desire to attract different consumer segments with varying needs and purchasing behaviors. The adoption of four differentiation strategies is less common, found in only six sectoral brands (18.2%): South Africa, Switzerland, Georgia, Moldova, Portugal, and Hungary. The adoption of multiple differentiation strategies may indicate a higher level of sophistication and strong adaptability to the diverse demands of the global wine market. Among the analyzed sectoral brands, only three—Argentina, Greece, and Lebanon (9.1%)—utilize five differentiation strategies, suggesting a highly diversified marketing approach. The simultaneous application of multiple differentiation strategies can create confusion in wine consumers’ perceptions. To prevent such undesirable effects and to optimize resource allocation, it is essential for sectoral brands to identify those differentiation strategies—or combinations thereof—that have the greatest impact in relation to their specific objectives [61,62].

### 4.1. Differentiation Strategies and Attributes Allocation

The most frequently adopted differentiation strategies by sectoral brands in the global wine industry are presented in the following paragraphs (Figure 3).

The differentiation strategy through an “attribute” is used by 25 sectoral brands (75.8% of the total). This is the most frequently employed strategy, suggesting that most countries consider specific elements of their national wine industry as essential for distinguishing themselves in the global wine market. The differentiation strategy through “tradition” is used by 17 sectoral brands (51.5% of the total). Tradition is a powerful factor in wine promotion, reflecting that many countries build their sectoral brand around their viticulture heritage and historical winemaking practices [63]. The differentiation strategy through “how it is made” (production method) is adopted by 13 sectoral brands (39.4% of the total). These brands emphasize their unique winemaking processes to attract consumers who value transparency and different (modern) winemaking technologies. The differentiation strategy through “heritage” (used by 10 sectoral brands, 30.3% of the total) allows these national industries to present themselves as having a strong historical and cultural connection to winemaking. This can be appealing to consumers interested in authentic stories and the origins of wine [64].

“Leadership” is adopted as a differentiation strategy by six sectoral brands (18.2% of the total). This strategy focuses on industry leadership, whether in quality, quantity, or innovation. The countries using this strategy position themselves at the forefront of the wine industry, as leaders in production or winemaking technologies. The differentiation strategy through “being a specialist” is used by six sectoral brands (18.2% of the total). This suggests that these countries focus on producing a specific type of wine or catering to a particular niche market, aiming to be perceived as experts in a certain variety of wine or production method [65]. The differentiation strategy through “being the latest” was chosen by five sectoral brands (15.2% of the total). This strategy involves the desire to be perceived as a “new” or “latest” brand in the market, bringing innovation or a fresh perspective to the industry. It also signals an intention to attract consumers through originality and modern features in both the wine itself and the production process. The differentiation strategy through “consumer group preference” (used by four sectoral brands, 12.1% of the total) is based on targeting a specific consumer segment or promoting a national wine industry as the preferred choice of a particular group (community, age group, or consumers with specific preferences). The differentiation strategy through “being the first”, adopted by two sectoral brands (6.1% of the total), focuses on pioneering the entire industry (or a segment of it). This is a less commonly used approach, but it can signify innovation and leadership, which may influence consumer purchasing decisions.

Because the differentiation strategy through an “attribute” is the most commonly used, the research further investigated how sectoral brands in the global wine industry implement this approach. A total of 27 distinct attributes were identified (Annex 1). The number of distinct attributes used by each sectoral brand in their differentiation strategy is presented in Figure 4.

The sectoral brands with the highest number of attributes used (seven) are those of Argentina, Australia, and Austria (25.9% of the total attributes). These sectoral brands are the most active in using attributes to differentiate their national wine production. The use of seven distinct attributes suggests either a complex, diversified approach aimed at attracting a broad range of consumers or an inability to identify a single unique and relevant attribute that would make the sectoral brand truly distinct and memorable. The sectoral brands that use three attributes in their differentiation strategy are South Africa, Brazil, Greece, North Macedonia, New Zealand, and Portugal (accounting for 11.1% of the total attributes). These countries have a more balanced approach, offering significant diversity in differentiation, though perhaps not as complex as the top three countries.

Chile, Croatia, Great Britain, Spain, Turkey, Ukraine, Hungary, and Uruguay are the countries whose sectoral brands rely on two attributes in their differentiation strategy (7.4% of the total attributes). These countries focus their differentiation strategy on highlighting the key aspects essential to their brand, aiming to attract a specific audience. The sectoral brands using only one attribute in their differentiation strategy are Belgium, Canada, China, Georgia, Germany, Israel, and Lebanon (3.7% of the total attributes). These countries emphasize a single distinctive aspect that is essential to their differentiation strategy.

Regarding the attributes used and how they are applied by sectoral brands in their differentiation strategies, the detailed situation is presented in the paragraphs below (Figure 5). 

The most frequently used attribute is “terroir” (11 countries, 44% of the total), suggesting that many of the analyzed sectoral brands place significant emphasis on origin and local land characteristics (climate, soil, geographic factors, etc.) to differentiate their national wine production. “Terroir” is an important concept in the wine world, highlighting the strong connection between wine and its production environment [66].

Other frequently used attributes, though with a lower share, include “quality” and “diversity” (six countries, 24% of the total), “sustainable”, “indigenous grapes”, and “friendly” (four countries, 16% of the total). “Quality” and “diversity” appear to be essential attributes for many wine-producing countries that choose to focus on these aspects to attract consumers. “Quality” suggests superior wine, while “diversity” implies a broad range of wines, catering to different tastes and occasions. The other attributes reflect an interest in sustainable production, the promotion of specific local grape varieties to attract wine enthusiasts, and the creation of a friendly brand image that appeals to consumers through a message of accessibility and warmth [67].

Attributes with a lower frequency of use but significance in differentiation strategies include “fresh” (three countries, 12% of the total), “fun” and “pure” (two countries, 8% of the total). These attributes can be associated with a refreshing and enjoyable consumption experience, appealing to consumers who prefer easy-to-drink wines with a clean and vibrant taste profile. Attributes used only once (4% of the total) include “adventurous,” “quality & price”, “comfortable”, “conscious”, “exciting”, “fast-growing”, “reliable”, “elegant”, “innovative”, “modern”, “unchanged”, “untamed”, “for every moment”, “hospitable”, “passionate”, and “personal”. These attributes can be considered specific and specialized, helping to position sectoral brands distinctly in consumers’ minds.

### 4.2. Clusters Based on Differentiation Strategies and Attributes

The next phase of the research aimed to: (1) analyze how each sectoral brand in the global wine industry relates to others based on the differentiation strategies used, and (2) examine how sectoral brands that differentiate through an “attribute” relate to others. To achieve this, a cluster analysis was conducted.

For the clusters focusing on sectoral brands based on the differentiation strategy used, the graphical representation of the results is shown in Figure 6 and Figure 7.

Cluster 1 includes the sectoral brands from the following countries: Australia, Austria, Brazil, Chile, China, Lichtenstein, Spain, and Uruguay. Within this cluster, the dominant differentiation strategy is through an “attribute” (Spain, Chile, Australia, Brazil, Austria, China, and Uruguay). The “heritage” differentiation strategy is used only in Chile, suggesting that its sectoral brand differentiates itself also through tradition and history. The “tradition” strategy appears in Lichtenstein, indicating a historically oriented approach, while “how it is made” is applied only in Austria, emphasizing the importance of the production process. The “Being the last” strategy is exclusive to Australia and Brazil, potentially indicating that these sectoral brands position themselves either as market newcomers or as leaders in modern technology.

Cluster 2 includes the sectoral brands of Armenia and Cyprus. In this case, both countries differentiate themselves through “tradition” and “heritage”, suggesting a strong focus on historical and cultural values as well as continuity in wine production. Cluster 3 consists of the sectoral brands of South Africa, Canada, Switzerland, Great Britain, and Portugal. In this group, differentiation through “attribute” and “leadership” are the dominant strategies used across all five countries, indicating that these brands stand out through product characteristics and market leadership positioning. “How it is made” is a strategy applied only in Switzerland and Canada, suggesting that in these countries, quality and the production process are key differentiation factors. “Tradition” is used only in Portugal, highlighting a focus on history and continuity in branding, and Portugal is also the only country that uses “specialist” differentiation, indicating a niche market strategy positioning the brand as an expert in wine production. “Heritage” is a strategy exclusively used in South Africa, implying that the country’s brand places value on its past and long-standing traditions. “Group preference” appears only in South Africa and Switzerland, suggesting that sectoral brands in these countries are particularly appreciated or consumed by specific consumer segments.

Cluster 4 includes the sectoral brands of Germany, Lithuania, New Zealand, and Slovenia. All four countries emphasize quality and the production process (“how it is made”), reflecting a technical and excellence-driven approach. Germany and New Zealand use both “attribute” and “how it is made”, indicating that brands in these countries differentiate themselves through both tangible product characteristics and the production process. Slovenia focuses on tradition and specialization, indicating a more niche approach, while New Zealand applies the “being the last” strategy, possibly positioning itself as a new or emerging player in the market and a leader in technological advancements in winemaking. Lithuania has the narrowest differentiation strategy, relying solely on “how it is made”, which may suggest a less diversified industry or a strict focus on quality and production methods. 

Cluster 5 includes the sectoral brands from Argentina, Greece, Israel, and Turkey. Here, the focus is on product characteristics (“attribute-based differentiation”) and the production process (“how it is made”), indicating an industry driven by quality and innovation. Argentina and Greece use the most diverse strategies, including “tradition” and “heritage”, suggesting that sectoral brands in these countries leverage history and culture. Turkey also differentiates itself through “being the last”, which may indicate a modern and innovative positioning. Greece is the only country that employs a niche (“specialist”) strategy, while Israel combines tradition with innovation but does not use “heritage” as a differentiation strategy, suggesting a more pragmatic focus on modern values while maintaining historical roots.

Cluster 6 consists of the sectoral brands from Belgium, Bulgaria, Estonia, Croatia, Georgia, Lebanon, North Macedonia, and Ukraine. In this case, the sectoral brands from Georgia, Croatia, North Macedonia, Ukraine, and Lebanon emphasize “tradition” and “heritage”, indicating industries where history and cultural values play an important role in consumer relations. Belgium and Estonia are the only countries that differentiate themselves through “how it is made”, suggesting industries focused on technology and quality. Lebanon also applies the “specialist” and “being the first” strategies, while Bulgaria has a limited strategy based only on “tradition”. 

Cluster 7 includes the sectoral brands from Moldova and Hungary. Both countries emphasize “tradition”, indicating a strong connection with cultural values and local history in their differentiation strategies. At the same time, they also differentiate themselves through “specialist” positioning, suggesting a niche strategy and a focus on excellence. “Group preference” is another common strategy, aimed at attracting and retaining specific consumer segments. Differentiation through “attribute” is used only by Hungary, while Moldova is the only country that applies “leadership” as a strategy, implying that its sectoral brand seeks to be perceived as a market leader or an innovator in the field.

For clusters focusing on sectoral brands based on the attributes used for differentiation strategies, the graphical representation is shown in Figure 8 and Figure 9.

Cluster 1 includes the sectoral brands from Brazil, Chile, Croatia, Greece, Macedonia, Great Britain, New Zealand, Spain, and Ukraine. The sectoral brands from these countries adopt varied differentiation strategies based on attributes, with some focusing on quality and taste (Ukraine, Great Britain, Croatia), while others emphasize purity and unchanged (Chile, New Zealand, Macedonia). Brazil stands out with a friendly, fun, and accessible image, while Spain focuses on social and consumption experiences. Macedonia and Chile promote their products through authenticity, highlighting the unaltered characteristics of their wines, whereas New Zealand and Great Britain suggest innovation and sustainability, indicating a strong forward-thinking orientation.

Cluster 2 consists of sectoral brands from South Africa, Belgium, Canada, China, Georgia, Germany, Israel, Lebanon, Portugal, Turkey, Hungary, and Uruguay. Most of these countries emphasize terroir and winemaking traditions, suggesting a strong association with the wine industry and regional authenticity. Belgium and Uruguay stand out for their focus on quality, indicating an emphasis on product excellence. South Africa and Germany build their identity through diversity, while China differentiates itself through freshness, which may suggest an industry oriented towards new and innovative products.

Cluster 3 includes the sectoral brands from Argentina, Australia, and Austria. Argentina associates itself with attributes such as diversity, friendliness, joy, passion, sustainability, and terroir. Australia aims to be perceived as adventurous, environmentally conscious, friendly, and trustworthy, emphasizing quality and price, sustainability, and terroir. Austria focuses on attributes such as convenience, elegance, and hospitality, projecting an image of a high-quality, sustainable wine industry based on terroir, with a welcoming and enjoyable atmosphere.

## 5. Conclusions

### 5.1. Implications

The data presented in this research indicate that the majority of sectoral brands in the wine industry (51.5%) choose to use two or three differentiation strategies, suggesting a balance between diversification and focus. Using a single strategy may either indicate a strong focus on a specific segment or limited resources in developing other differentiation directions.

The most commonly used differentiation strategies are through an “attribute” (75.8% of the analyzed sectoral brands) and “tradition” (51.5%), which suggests that most countries focus on the unique characteristics of their products/industry and on winemaking traditions to stand out. Strategies such as “being the first” and “consumer group preference” are less frequently used, indicating that they are more niche or specific approaches.

There is a clear trend toward diversifying differentiation strategies, but the risk for sectoral brands that employ too many strategies is that they may create confusion among consumers, ultimately failing to establish a clear and unique positioning in their minds. Managers should evaluate the optimal number of differentiation strategies employed, balancing the necessity to appeal to multiple consumer segments while maintaining a clear and consistent brand message. From a theoretical perspective, this study confirms and enlarges the current comprehension of differentiation strategies by integrating them within sectoral branding theory, particularly emphasizing the dynamic interaction between market positioning and brand attributes.

Differentiation through an “attribute” is the most widely used strategy, and the study reveals that the highest number of countries (eight) use two differentiation attributes, followed by seven countries that use only one attribute. “Terroir” is the most commonly used attribute, highlighting the importance of origins and connection to the natural environment in the wine industry. “Quality” and “diversity” are commonly emphasized attributes, reflecting a commitment to excellence and a broad product range. Meanwhile, the attributes “adventurous” and “innovative” are used to appeal to market segments looking for new experiences and a contemporary approach to winemaking.

Overall, this diversification of attributes suggests that wine differentiation strategies are highly varied, catering to different consumer groups—from those seeking tradition and quality to those drawn to innovation and sustainable products. While this approach may be justified by the attempt to target multiple markets, it is highly likely that it creates confusion for consumers. In an overcrowded global market filled with information, the key to success lies in delivering simple, unique, and relevant messages.

The people that build new wine market policies could consider supporting sectoral branding initiatives at a national or regional level, particularly those focusing on unique attributes like terroir, indigenous grape varieties, or sustainability, and in that way facilitating stronger international wine market positions. Leveraging digital marketing channels and social media might amplify the impact of differentiation strategies, allowing sectoral brands to clearly communicate their distinctive attributes to consumers.

For countries that have not yet developed a sectoral brand for their wine industry, these conclusions can also be seen as opportunities to explore less commonly used differentiation strategies such as “leadership”, “specialist”, “being the last”, and “consumer group preference”. Additionally, developing sub-brands for wine regions and the most representative local grape varieties could help create a stronger and more distinct positioning.

### 5.2. Limitations and Future Research Directions

This study presents several limitations. The most significant is the exclusive use of differentiation strategies (methods) proposed by Kotler and Keller [24]. Another limitation concerns the interpretation given by the authors to key terms, attributes, and slogans identified on websites and promotional materials used by sectoral brands in their differentiation strategies. This interpretation formed the basis for the classifications in this study.

Even though it was not the primary objective, the research highlighted that, with very few exceptions (Hungary, Croatia, Switzerland, and Portugal), no other sectoral brand has developed sub-brands, co-brands, or endorsed brands to promote the wine regions of their respective countries. Additionally, very few sectoral brands (with exceptions such as Argentina with Malbec, Portugal with Vinho Verde, and Hungary with Tokaji Aszú) have created sub-brands for an indigenous grape variety used in winemaking or a signature wine that represents the country [68].

From a methodological point of view, future studies could benefit from employing mixed-method approaches, combining cluster analyses with qualitative consumer insights or surveys, to validate findings and provide deeper consumer-oriented perspectives [69].

The future research directions considered by the authors include examining consumer perspectives, both at the global wine market level and within specific geographic segments, on: which of the differentiation strategies identified in this study are relevant to the purchasing decision; which of the differentiation attributes identified in this study influence consumer choice; and developing positioning maps of sectoral brands based on consumer perceptions and identifying positioning gaps between the differentiation strategy used and consumer perceptions of it. Supplementary, future studies should analyze wine consumer responses to differentiation strategies in digital environments, examining how digital marketing influences consumer perceptions and decision-making processes regarding sectoral wine brands.

Also, from the perspective of geographical expansion, further comparative research could explore differentiation strategies across established and emerging wine markets, identifying similarities and differences in effectiveness, thus helping practitioners adapt their strategies to specific regional contexts. Another direction of future research will focus on brand architecture for sectoral brands in the wine industry, exploring how they can accommodate a portfolio of five or more distinct brands, covering wine regions, vineyards, indigenous grape varieties, producer brands, and the wine itself [70,71,72].

Last but not least, in response to the growing wine consumer preference for sustainability, future studies could examine how differentiation strategies centered on sustainability influence wine consumer decisions and have an impact on wine brand performance within the world wine market.

## Figures and Tables

**Figure 1 foods-14-01858-f001:**
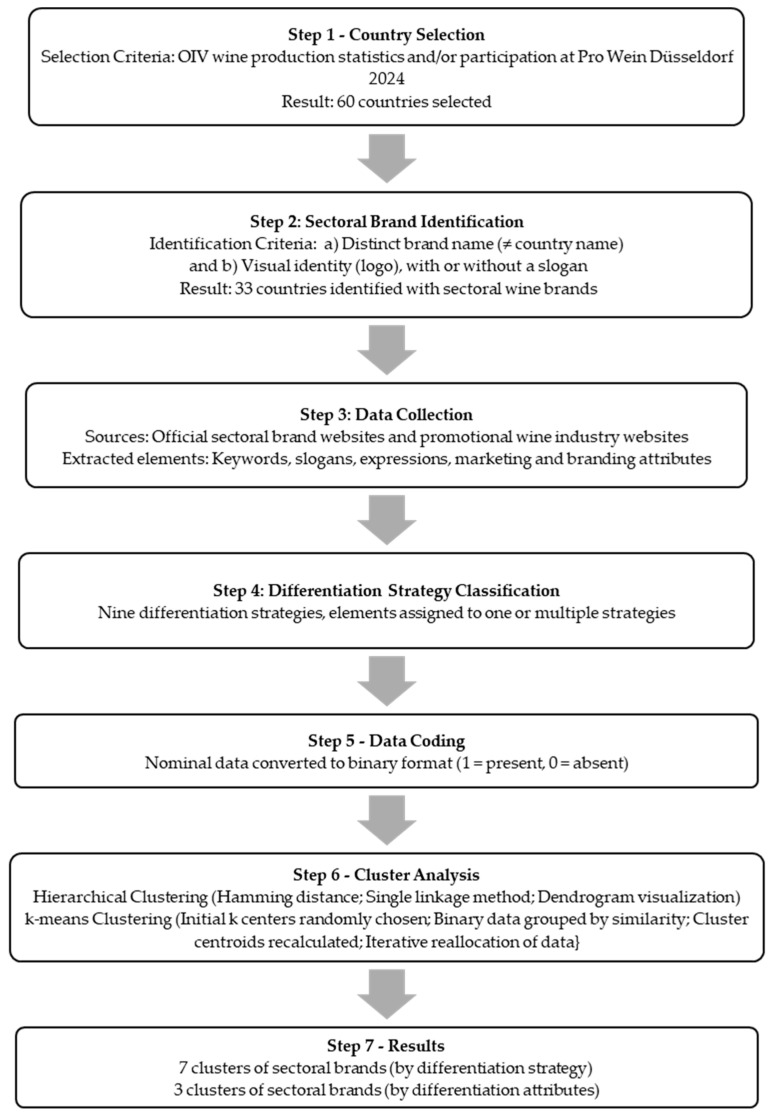
Steps of the research methodology.

**Figure 2 foods-14-01858-f002:**
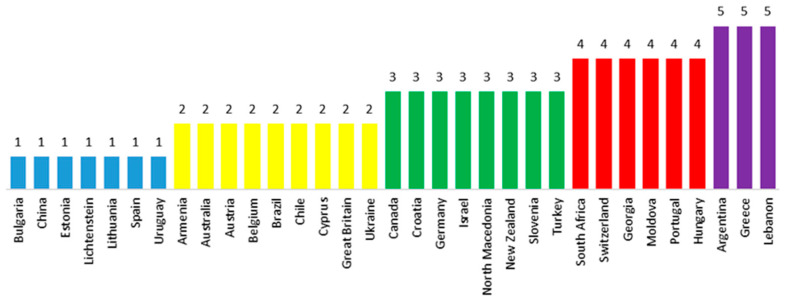
Number of differentiation strategies used by sectoral brands in the global wine industry.

**Figure 3 foods-14-01858-f003:**
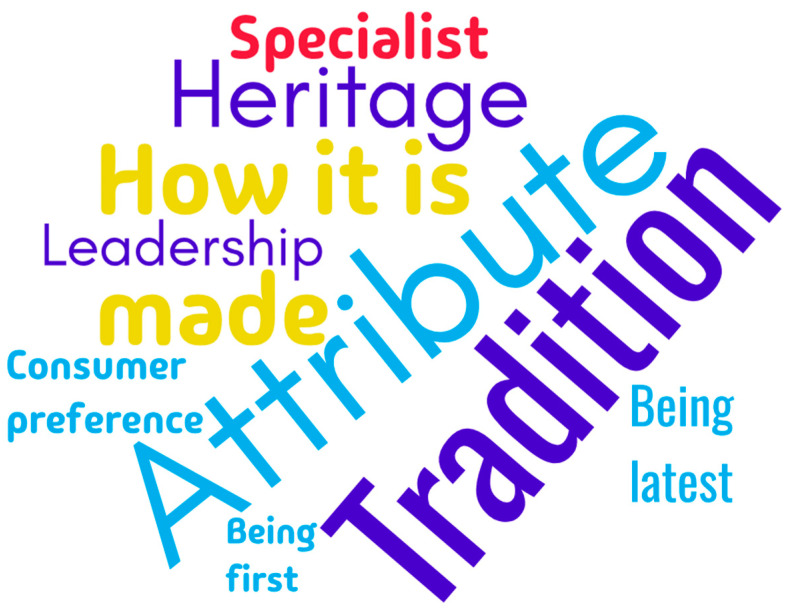
The most used differentiation strategies.

**Figure 4 foods-14-01858-f004:**
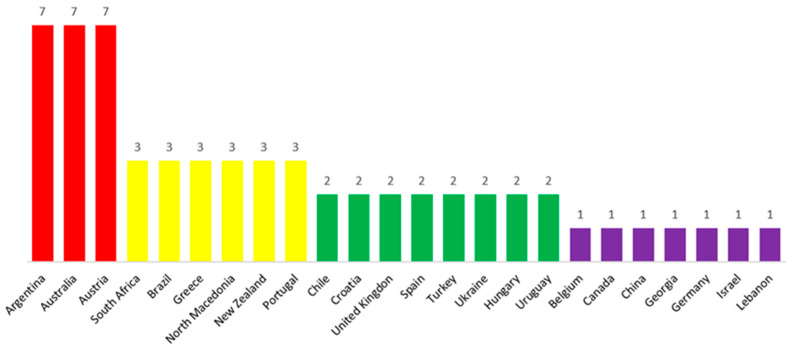
Number of distinct attributes used in the differentiation strategy.

**Figure 5 foods-14-01858-f005:**
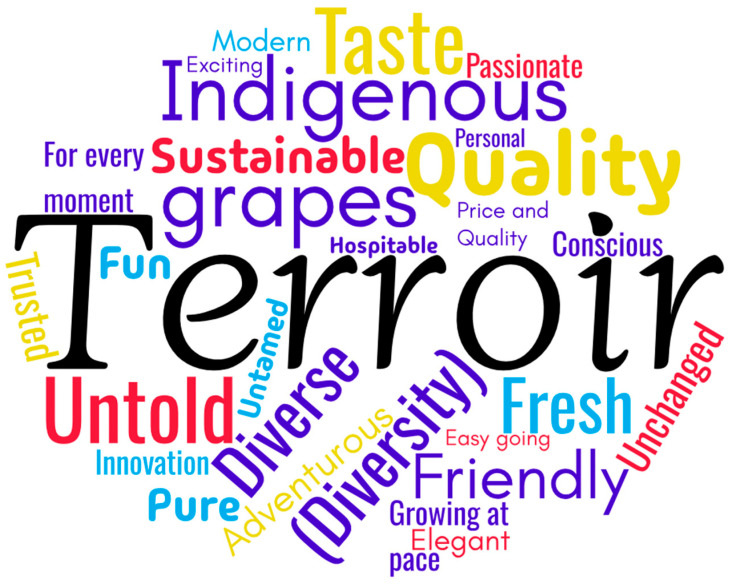
Attributes used in differentiation strategies.

**Figure 6 foods-14-01858-f006:**
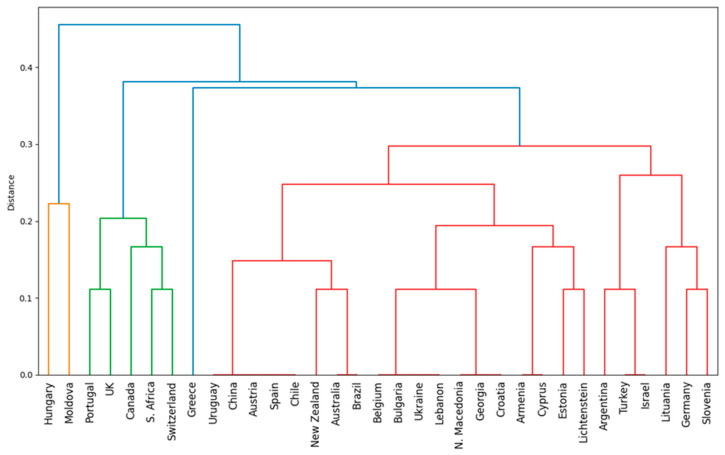
The hierarchical clustering dendrogram of sectoral brands based on the differentiation strategy used.

**Figure 7 foods-14-01858-f007:**
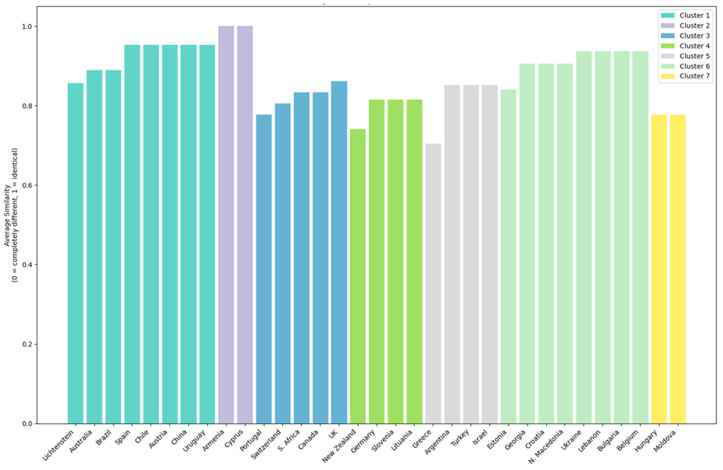
Average similarity to cluster members based on the differentiation strategy used.

**Figure 8 foods-14-01858-f008:**
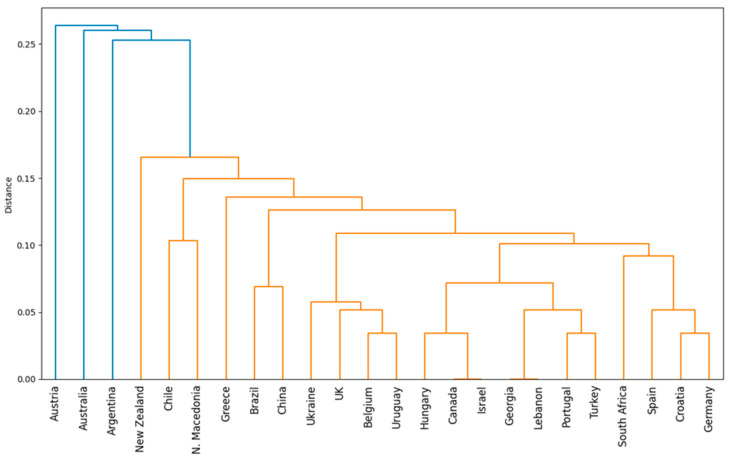
The hierarchical clustering dendrogram of sectoral brands based on the differentiation attributes used.

**Figure 9 foods-14-01858-f009:**
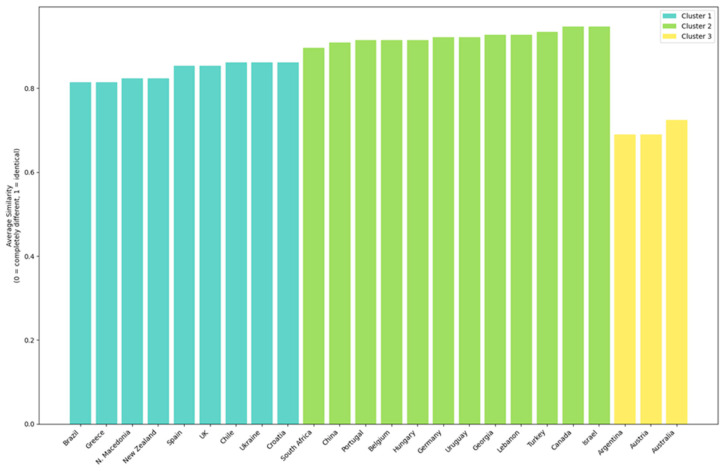
Average similarity to cluster members based on differentiation attributes used.

## Data Availability

The original contributions presented in the study are included in the article, further inquiries can be directed to the corresponding author.

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
