# Peer review of "Competitive Advantage in the World of Wine—An Analysis of Differentiation Strategies Developed by Sectoral Brands in the Global Market"

_foods, 2025, doi:10.3390/foods14111858_

Round 1
Reviewer 1 Report
Comments and Suggestions for Authors
In this paper, the authors analyzed the differentiation strategies developed by sectoral brands in the global wine industry and how these strategies interrelate. This topic is interesting. Overall, the data collections, the figures and table, the results and discussions were reasonable. I would like to recommend it for publication after minor revisions. The detailed comments are as follows:
- Line 37, The abbreviation ‘EU’ was used in first time in this article. The formal name must be indicated. Also Line 278 OIV.
- Are there any other studies related to this topic? Should be explained clearly in the end of Introduction, to highlight the innovation and importance of this study.
- The references are mainly in Chapter 1, 2, 3. Add some references in 4 and 5 because Results and discussion are important.
- 5 Conclusions maybe revised to 5 Discussion.
- Appendix maybe provided in supplementary materials.
- Check the references styles. For example, line 719, no volume and page numbers.
Author Response
Dear reviewer,
Thank you for your valuable feedback and suggestions. We deeply appreciate your help. The points you have raised have been addressed as follows:
Comments 1: Line 37, The abbreviation ‘EU’ was used in first time in this article. The formal name must be indicated. Also Line 278 OIV.
Response 1: We changed from EU to European Union (EU) (line 39) and from OIV to Organisation Internationale de la Vigne et du Vin (OIV) (line 300).
Comments 2: Are there any other studies related to this topic? Should be explained clearly in the end of Introduction, to highlight the innovation and importance of this study.
Response 2: We have introduced a new paragraph: “The importance of the country of origin in wine trade is a subject that has been extensively analyzed both in academic literature and in current commercial practice. However, to date, there is a lack of research examining how sectoral brands within the global wine industry formulate their differentiation strategies, taking into account both their own resources and the strategic decisions of their competitors”. (lines 74-78).
Comments 3: The references are mainly in Chapter 1, 2, 3. Add some references in 4 and 5 because Results and discussion are important.
Response 3: Following your suggestions, we added 9 new references in chapters 4 and 5 ([61]-[67], [69] and [72].
Comments 4: 5 Conclusions maybe revised to 5 Discussion.
Response 4: We have taken into account all the comments made by you and the other reviewers and have decided to rename Chapter 4 to 'Results and Discussion' and introduced two subtitles (4.1. Differentiation strategies and attributes allocation (line 414) and 4.2. Clusters based on differentiation strategies and attributes (line 517)).
Comments 5: Appendix maybe provided in supplementary materials.
Response 5: We consider the information presented in the Annex to be easier to correlate, in this format, with the arguments made in the article’s chapters, which is why we would prefer to maintain the current structure. However, if you consider it absolutely necessary for this information to appear in the supplementary materials, we can make that adjustment
Comments 6: Check the references styles. For example, line 719, no volume and page numbers.
Response 6a: For the paper “Keskin, H., Senturk, H.A., Tatoglu, E., Golgeci, I., Kalaycioglu, O., Etlioglu, T. The Simultaneous Effect of Firm Capabilities and Competitive Strategies on Export Performance: The Role of Competitive Advantages and Competitive Intensity. International Marketing Review, 2021” – this is the way the publication recommends citation (https://pure.au.dk/portal/en/publications/the-simultaneous-effect-of-firm-capabilities-and-competitive-stra ) (line 763)
Response 6b: For the paper “Dressler, M.; Paunovic, I. The Value of Consistency: Portfolio Labelling Strategies and Impact on Winery Brand Equity. Sustainability” - we have updated the reference (line 768).
Reviewer 2 Report
Comments and Suggestions for Authors
I am grateful for the opportunity to review a quality manuscript that deals with a topic that I have been studying for many years. I will give some suggestions for small changes, which can certainly contribute to the quality of the manuscript, but in no way diminish it, because I express all praise for the first reading of the manuscript.
Namely, the manuscript deals with the current and relevant topic of the use of fermented and preserved plant products in modern nutrition, combining chemical, nutritional and sensory aspects. The title is adequate and clear. The abstract contains all the necessary elements, no changes are necessary. The introduction is comprehensively conceived, but I am of the opinion that the introduction should end with a clearer highlighting or identifying a gap in the literature and setting the research goal. Also, in the introductory part, authors can ask research questions, but it is not necessary. I leave them the freedom to decide for themselves in accordance with their idea and the reviews of other reviewers. The literature review is very extensive and contains all the necessary elements with which the authors cover the essence and meaning of their research. I would like to note that they would only slightly strengthen the critical approach to all findings. The methodology is clear and no major changes are needed. My suggestion is that the authors should further clarify how the product samples were selected and how the storage conditions were controlled. The results are very clearly and legibly presented in tables and figures, but part of the text is very often repeated, so I ask the authors to check the text to read it completely and not repeat sentences. The discussion sometimes deviates from the results and is more like a literature review. It is necessary to connect the findings more strongly with previous studies and offer a deeper interpretation of the results. The authors clearly stated the concluding considerations, significance and implications, but they also stated the real limitations of the research and directions for future research. The references are adequate.
Author Response
Comments 1: I am grateful for the opportunity to review a quality manuscript that deals with a topic that I have been studying for many years. I will give some suggestions for small changes, which can certainly contribute to the quality of the manuscript, but in no way diminish it, because I express all praise for the first reading of the manuscript.
Namely, the manuscript deals with the current and relevant topic of the use of fermented and preserved plant products in modern nutrition, combining chemical, nutritional and sensory aspects. The title is adequate and clear. The abstract contains all the necessary elements, no changes are necessary.
The introduction is comprehensively conceived, but I am of the opinion that the introduction should end with a clearer highlighting or identifying a gap in the literature and setting the research goal. Also, in the introductory part, authors can ask research questions, but it is not necessary. I leave them the freedom to decide for themselves in accordance with their idea and the reviews of other reviewers.
The literature review is very extensive and contains all the necessary elements with which the authors cover the essence and meaning of their research. I would like to note that they would only slightly strengthen the critical approach to all findings.
The methodology is clear and no major changes are needed. My suggestion is that the authors should further clarify how the product samples were selected and how the storage conditions were controlled.
The results are very clearly and legibly presented in tables and figures, but part of the text is very often repeated, so I ask the authors to check the text to read it completely and not repeat sentences. The discussion sometimes deviates from the results and is more like a literature review. It is necessary to connect the findings more strongly with previous studies and offer a deeper interpretation of the results.
The authors clearly stated the concluding considerations, significance and implications, but they also stated the real limitations of the research and directions for future research. The references are adequate.
Response 1:
Dear reviewer,
Thank you very much for your analysis and the valuable observations you provided. We will take them into account and incorporate them into our future research endeavors.
Reviewer 3 Report
Comments and Suggestions for Authors
Authors conduct analyses on the differentiation strategies and the correlations between them. Overall, the words of this article is not concise enough and the structure needs to be adjusted. Some revisions are needed as follows.
- The title is inappropriate. It does not fully reflect the research contents of this article. Don’t repeat the same words in the title.
- The citation of references in the text is not standardized.
- At the end of Introduction, authors can directly introduce what researches this article are going to do, there is no need to list the main content of the following chapters.
- In Literature Review, please add a paragraph describing the correlation between the subsequent sections and this article.
- In Methodology, please provide a flowchart to illustrate the approach.
- The Results section can be revised as Results and Discussion. Please add some subtitles in this section.
- In Figure 4 and Figure 6, the main differences are reflected in the upper part of the bar chart, and the values in the lower part can be removed without starting from 0.0.
Author Response
Dear reviewer,
Thank you for your valuable feedback and suggestions. We deeply appreciate your help. The points you have raised have been addressed as follows:
Comments 1: The title is inappropriate. It does not fully reflect the research contents of this article. Don’t repeat the same words in the title.
Response1: Following the observation received, we have reformulated the title of the article to “Competitive Advantage in the World of Wine – An Analysis of Differentiation Strategies Developed by Sectoral Brands in the Global Market” (lines 2-6).
Comments 2: The citation of references in the text is not standardized.
Response2: We have revised the referencing format in the text and hope that all inconsistencies have been resolved.
Comments 3: At the end of Introduction, authors can directly introduce what researches this article are going to do, there is no need to list the main content of the following chapters.
Response 3: We have reformulated the final paragraph of the Introduction section as follows: “This study investigates how differentiation strategies contribute to building competitive advantage for sectoral brands within the global wine industry. Drawing on existing theoretical frameworks, the research develops a methodology to analyze brand differentiation strategies. The findings offer practical implications for wine-producing countries, marketing and brand managers, and policymakers, highlighting effective approaches to strengthen market presence through targeted differentiation efforts” (lines 96-101).
Comments 4: In Literature Review, please add a paragraph describing the correlation between the subsequent sections and this article.
Response 4: Following the recommendations received, we have added the following paragraph – “The structure of the study reflects a coherent progression from conceptual grounding to practical application. The definition of sectoral brands establishes the foundation for the analysis, which is then operationalized through a mixed-methods approach involving both quantitative and qualitative techniques. This methodological framework supports the identification of strategic and attribute-based clusters, revealing patterns of differentiation among the examined sectoral brands. The integration of cluster analysis not only facilitates the interpretation of complex relationships but also bridges the gap in literature by offering a comprehensive perspective on differentiation in the global wine industry. Together, these sections contribute to a unified narrative that advances both academic understanding and practical strategy development (lines 279-288).”
Comments 5: In Methodology, please provide a flowchart to illustrate the approach.
Response 5: A flowchart highlighting the steps of the research methodology has been added (lines 382-384).
Comments 6: The Results section can be revised as Results and Discussion. Please add some subtitles in this section.
Response 6: Following your suggestion, two subtitles have been introduced (4.1. Differentiation strategies and attributes allocation (line 414) and 4.2. Clusters based on differentiation strategies and attributes (line 517)).
Comments 7: In Figure 4 and Figure 6, the main differences are reflected in the upper part of the bar chart, and the values in the lower part can be removed without starting from 0.0.
Response 7: Figures 4 and 6 (which became Figures 7 and 9 after the addition of the flowchart and others figures based on Table 2 and 3) are automatically generated by the software used for the statistical operations and its libraries (Python 3.8.10, SciPy, Scikit-learn, Matplotlib, and Seaborn), and we are unable to alter the graphical representation of these data.
Reviewer 4 Report
Comments and Suggestions for Authors
Manuscript ID: foods--3604467
Title: Differentiation in the World of Wine – An Analysis of Differentiation Strategies Developed by Sectoral Brands in the Global Wine Industry
The study explores the differentiation strategies in the world wine market and their relationships. The cluster analysis approach unveiled relationships between sectoral wine brands based on their differentiation strategies. The study's practical significance lies in identifying opportunities for wine-producing countries that have not yet developed sectoral brands.
The study's topic is relevant to food science and fits the journal's scope. The study is well-structured and neatly written. The article's abstract clearly and accurately reflects the study. The introductory and literature review sections provide an appropriate research background. The authors address a well-defined research task (identification and interpretation of the differentiation strategies of sectoral brands in the global wine industry), employing sound and technically solid methodology (developing a database with sixty countries, based on their OIV wine production statistics and/or their participation at the world’s most important wine trade fair; identification of a sectoral brand for national wine industry (thirty-three sectoral brands developed by wine-producing countries); analysis of official sectoral brand websites or websites that promote the wine industry – extraction of keywords, expressions, slogans, and other marketing and branding elements to identify differentiation strategies and the attributes used for differentiation, based on previously established identification criteria and further used for grouping). The work is described in sufficient detail. The statistical analyses are appropriate and well reported (a cluster analysis approach - hierarchical clustering algorithm, K-means algorithm). The work and obtained results are aligned with the research aim, documented and interpreted consistently and correctly throughout the article. The conclusions are consistent with the evidence presented. The study limitation section is excellently combined with indications of future research directions. Cited references are appropriate. Data is contained within the article.
Comments:
Lines 319-320: Table 1 does not add value to the manuscript and should be deleted. A binary coding scheme (0 / 1) can easily be added in line 317 (...binary values (0 / 1)).
Line 355, the whole Results section: This Chapter should be named Results and Discussion. However, for that purpose, it should provide context by comparing/commenting on the outcomes of published studies dealing with corresponding strategies and attributes used in the wine industry/marketing.
Line 382 – Table 2, and Lines 384-416: data presented in Table 2 and the mentioned text paragraph are different (different values are attributed to the same differentiating strategy). The authors must check and correct the data.
Line 282: Table 2 does not add value to the manuscript and should be deleted (all values are cited in the text). For visualisation purposes, the authors should opt for another approach, for example, Word Cloud (a freely available internet tool).
Line 447: Table 3 does not add value to the manuscript and should be deleted (all values are cited in the text, except information that 1 usage corresponds to 4%, which can easily be added in line 468 (Attributes used only once (4%)...). For visualisation purposes, the authors should opt for another approach, for example, Word Cloud (a freely available internet tool).
Lines 454-457: to facilitate readability and fast recognition, avoid unnecessary repetition. ...“quality” (6 countries, 24% of the total), “diversity” (6 countries, 24% of the total), “sustainable” (4 countries, 16% of the total), “indigenous grapes” (4 countries, 16% of the total), and “friendly” (4 countries, 16% of the total) = “quality” and “diversity” (6 countries, 24%, each); “sustainable”, “indigenous grapes” and “friendly” (4 countries, 16%, each).
Lines 465-466: the same as previous. ...“fresh” (3 countries, 12% of the total), “fun” (2 countries, 8% of the total), and 465 “pure” (2 countries, 8% of the total) = ...“fresh” (3 countries, 12% of the total), “fun” and “pure” (2 countries, 8%, each).
Author Response
Dear reviewer,
Thank you for your valuable feedback and suggestions. We deeply appreciate your help. The points you have raised have been addressed as follows:
Comments 1: Lines 319-320: Table 1 does not add value to the manuscript and should be deleted. A binary coding scheme (0 / 1) can easily be added in line 317 (...binary values (0 / 1)).
Response 1: Following your observation, we added the coding method to the text (line 344) and deleted table 1.
Comments 2: Line 355, the whole Results section: This Chapter should be named Results and Discussion. However, for that purpose, it should provide context by comparing/commenting on the outcomes of published studies dealing with corresponding strategies and attributes used in the wine industry/marketing.
Response 2: Following your comment, we renamed Chapter 4 to 'Results and Discussion' and introduced two subtitles (4.1. Differentiation strategies and attributes allocation (line 414) and 4.2. Clusters based on differentiation strategies and attributes (line 517)). However, throughout our literature review, we did not find any article that discusses the differentiation strategies and the attributes used by sectorial brands in the global wine industry.
Comments 3: Line 382 – Table 2, and Lines 384-416: data presented in Table 2 and the mentioned text paragraph are different (different values are attributed to the same differentiating strategy). The authors must check and correct the data.
Response 3: Thank you very much for your observation, we have made the necessary corrections in table 2 (lines 418-419).
Comments 4: Line 282: Table 2 does not add value to the manuscript and should be deleted (all values are cited in the text). For visualisation purposes, the authors should opt for another approach, for example, Word Cloud (a freely available internet tool).
Response 4: At your suggestion, we decided to eliminate table 2 and inserting an image generated with Word Cloud (lines 420-421).
Comments 5: Line 447: Table 3 does not add value to the manuscript and should be deleted (all values are cited in the text, except information that 1 usage corresponds to 4%, which can easily be added in line 468 (Attributes used only once (4%)...). For visualisation purposes, the authors should opt for another approach, for example, Word Cloud (a freely available internet tool).
Response 5: At your suggestion, we decided to eliminate table 3, add the sentence “4% of the total” (lines 511-512) and inserting an image generated with Word Cloud (lines 488-489).
Comments 6: Lines 454-457: to facilitate readability and fast recognition, avoid unnecessary repetition. ...“quality” (6 countries, 24% of the total), “diversity” (6 countries, 24% of the total), “sustainable” (4 countries, 16% of the total), “indigenous grapes” (4 countries, 16% of the total), and “friendly” (4 countries, 16% of the total) = “quality” and “diversity” (6 countries, 24%, each); “sustainable”, “indigenous grapes” and “friendly” (4 countries, 16%, each).
Response 6: Following the suggestion received, we have reformulated the respective paragraph (lines 496-499).
Comments 7: Lines 465-466: the same as previous. ...“fresh” (3 countries, 12% of the total), “fun” (2 countries, 8% of the total), and 465 “pure” (2 countries, 8% of the total) = ...“fresh” (3 countries, 12% of the total), “fun” and “pure” (2 countries, 8%, each).
Response 7: Following the suggestion received, we have reformulated the respective paragraph (lines 507-509).
Round 2
Reviewer 3 Report
Comments and Suggestions for Authors
It is recommended to accept and publish this paper.